genetics

AGCU Y37 PCR amplification kit, haplotype,
population structure, Guizhou Miao,
Guizhou Tujia

**Authors for correspondence:**
Yingnan Bian
e-mail: bianyingnan@yeah.net
Pengyu Chen
e-mail: pychenfs@163.com

†Li Luo and Lilan Yao contributed equally to this work.

# Forensic characteristics and population construction of two major minorities from southwest China revealed by a novel 37 Y-STR loci system

Li Luo[1,2,3,†], Lilan Yao[1,2,†], Siyu Chai[1,2], Hao Zhang[1,2], Min Li[3,4], Jian Yu[2], Xijie Hu[1,2], Chengtao Li[3,4], Yingnan Bian[3] and Pengyu Chen[1,2]

[1]Key Laboratory of Cell Engineering in Guizhou Province, Affiliated Hospital of Zunyi Medical University, Zunyi 563099, Guizhou, People's Republic of China
[2]School of Forensic Medicine, Zunyi Medical University, Zunyi 563099, Guizhou, People's Republic of China
[3]Shanghai Key Laboratory of Forensic Medicine, Shanghai Forensic Service Platform, Academy of Forensic Science, Shanghai 200063, People's Republic of China
[4]Institute of Forensic Medicine, West China School of Basic Medical Sciences and Forensic Medicine, Sichuan University, Chengdu 610000, Sichuan, People's Republic of China

LL, 0000-0002-2830-9814; YB, 0000-0002-8651-8737; PC, 0000-0002-3706-103X

Y-chromosome short tandem repeats (Y-STRs) have become important supplementary evidence in forensic science. Nowadays, the Y-chromosome STR haplotype reference database (YHRD) contains abundant Y-STR haplotype data from all over the world, while haplotype data of Guizhou Miao and Tujia are scarce. Hence, genetic polymorphisms of 37 Y-STRs were investigated in 446 unrelated males (206 Miao males and 246 Tujia males) residing in Guizhou Province. A total of 206 and 242 unique haplotypes with the highest diversity value of 0.9665 and 0.9470 were obtained. The heatmap, multidimensional scaling (MDS), the unweighted pair-group method with arithmetic means (UPGMA) tree and principal component analysis (PCA) based on the genetic distance (Rst) value within our studied populations and other 26 populations indicated that population structures follow the boundary of the continent. Guizhou Miao and Guizhou Tujia populations have intimate relationships with East Asian populations, especially the geographically close, similar history and the same language family populations.

# 1. Introduction

Y-chromosome short tandem repeats (Y-STRs), with the characteristics of male-specific, patroclinous and haplotype genetic, have been widely used in human evolution [1], genealogical research [2], population structure [3], forensic personal identification and paternity testing [4]. Therefore, the relevant databases are indispensable to ensure the haplotype frequencies estimated between two or more population-specific haplotypes [5,6]. Nowadays, the largest of freely accessible online database named the Y-chromosome STR haplotype reference database (YHRD, https://yhrd.org, release 62) have been established. The YHRD website contains abundant Y-STR haplotype data from diverse populations and ethnic groups all over the world. However, there are fewer population data reports of Y-STRs in Guizhou Miao and Guizhou Tujia populations. The AGCU Y37 PCR amplification kit contains six-colour fluorescence, which can greatly improve the polymorphic information content and individual discrimination (http://www.agcu.cn/page8?article_id=25&brd=1).

China, located in East Asia, is the world's most populous country. Additionally, China legally recognizes 56 distinct ethnic groups with a population of more than 1.4 billion in 2020 (https://en.jinzhao.wiki/wiki/China#cite_note-23). All ethnic groups comprise the Zhonghua Minzu. Miao and Tujia are the fourth and eighth largest minorities in China, which constitute approximately 0.7% and 0.6% of the total population (http://www.stats.gov.cn/tjsj/pcsj/rkpc/6rp/indexch.htm). Miao, also named Hmong, is an international ethnic group originated from China. At present, the Miao populations mainly live in southern China (Guizhou, Hunan, Sichuan, Yunan and so on) and some of the Miao populations have migrated out of China into Southeast Asia (Thailand, Laos and so on) (https://en.jinzhao.wiki/wiki/Miao_people). According to the sixth population census in 2010 (http://www.stats.gov.cn/tjsj/pcsj/rkpc/6rp/indexch.htm), the Miao populations are the largest ethnic group in Guizhou Province, with a population of 3.13 million. The Miao people have their own language (Hmong), which belongs to the Hmong-Mien language group of the Sino-Tibetan language family. In the early twentieth century, Miao people created their own writing system. Tujia is an ethnic group with a long history in China. Nowadays, the Tujia populations mainly reside in Guizhou Province, Hunan Province, Hubei Province and Chongqing Municipality bordering the Wuling mountains. Based on the sixth population census in 2010 (http://www.stats.gov.cn/tjsj/pcsj/rkpc/6rp/indexch.htm), about 1.02 million Tujia populations settled in Guizhou Province. The Tujia populations speak the Tujia language but have no script. The Tujia language is one of the Tibeto-Burman languages of the Sino-Tibetan language family (https://en.wikipedia.org/wiki/Tujia_people).

In the present study, 37 Y-STRs were typed to obtain the population data of the two populations. Moreover, the Y-chromosomal characteristics of AGCU Y37 PCR amplification kit were evaluated and the population structure with other populations from home and abroad were analysed. The geographical position is shown in electronic supplementary material, figure S1.

# 2. Material and methods

## 2.1. Population samples and ethical statement

Blood samples were obtained from 452 unrelated healthy male individuals who resided in Guizhou Province (southwest China). Among them, 206 Miao populations live in Kaili City, Eastern Guizhou, while the other 246 Tujia populations dwell at Tongren City, Northeast Guizhou. The inclusion criteria were as follows: (i) the volunteers were males; (ii) ancestors were Miao or Tujia populations; (iii) the people have been in Guizhou province for more than three generations. All volunteers provided informed consent under the approval of the Ethics Committee of the Zunyi Medical University (KLLY-2019-080).

## 2.2. DNA amplification and STR genotyping

Genomic DNA was separated by the salting-out method [7]. NanoDrop 2000c (Thermo Fisher Scientific) was used to assess the DNA concentration and extracted DNA was diluted to 2 ng $\mu l^{-1}$. The AGCU Y37 PCR amplification kit (AGCU ScienTech Incorporation, Wuxi, Jiangsu, China) was used to co-amplify 37 Y-STR loci (DYS392, DYS389I, DYS447, DYS389II, DYS438, DYS527a/b, DYS645, DYS596, DYS391, DYS456, DYS19, DYS593, DYS448, DYS449, DYS385a/b, DYS549, DYS437, DYS481, DYS533, DYS390, DYS627, DYS458, DYS460, DYS393, Y_GATA_H4, DYS439, DYS635, DYS444, DYS643, DYS557,

DYS576, DYS570, DYF387S1 and DYS518) on ProFlex $3 \times 32$-well PCR System (Applied Biosystems, Foster City, CA) following the manufacturer's protocol. A total of 10 µl PCR reaction volume were employed, including 4 µl of reaction mix, 2 µl of 37 Y-STR primers, 0.4 µl of Taq DNA polymerase, 1 µl of DNA template and 2.6 µl of sdH$_2$O sterile deionized H$_2$O (sdH$_2$O). PCR condition consisted of predegeneration of 2 min at 95°C; 30 cycles of 30 s at 94°C, 1 min at 60°C, 1 min at 72°C; final extension of 20 min at 60°C and hold at 4°C. The amplification products were detected and separated by ABI 3500XL Genetic Analyzer (Applied Biosystems, Foster City, CA). Finally, 37 Y-STR loci profiles were analysed using GeneMapper ID v. 1.4 (Applied Biosystems, Foster City, CA). A negative (ddH$_2$O) and a positive (DNA 9948) control run through each genotyping batch. Raw haplotype data of Guizhou Miao and Guizhou Tujia individuals were submitted to the YHRD and received the accession numbers of YA004671 and YA004672.

## 2.3. Data analysis

Haplotype and allele frequencies were carried out by direct counting. Three multi-copy loci (DYS527a/b, DYS385a/b and DYF387S1) were regarded as allelic combinations. Genetic diversity (GD) and haplotype diversity (HD) were computed by Nei's formula [8]: $n(1 - \sum pi^2)/(n-1)$, where $n$ means the sample size, and $pi$ denotes the frequency of the $i$th allele or haplotype. Match probability (MP) was determined as the sum of allele frequencies squared. Discrimination capacity (DC) was the ratio between the number of different haplotypes and the sample size. Twenty-eight populations from home and abroad were performed using analysis of molecular variance (AMOVA) [9] and multidimensional scaling (MDS) [10] tool on the YHRD website based on the same 27 Y-STRs (Yfiler Plus dataset). Pairwise genetic distances (Rst) were visualized by R Software v. 3.3 using the heatmap package. Unweighted pair-group method with arithmetic means (UPGMA) tree and principal component analysis (PCA) were constructed using Mega v. 7.0 [11] and SPSS v.26.0 [12].

# 3. Results

## 3.1. Population haplotype diversities of 37 Y-STR loci

Haplotypes containing 37 Y-STRs from 206 Miao male individuals and 246 Tujia male individuals residing in Guizhou Province were obtained (electronic supplementary material, tables S1 and S2). A total of 206 and 242 unique haplotypes were found, respectively. There were four different haplotypes with two repetitions in Tujia males. Allele frequencies and gene diversity values are presented in electronic supplementary material, tables S3 and S4. The allele numbers at single-copy loci ranged from three (DYS645) to 13 (DYS518) in Guizhou Miao individuals and varied from four (DYS389I, DYS437, DYS438 and DYS593) to 15 (DYS518) in Guizhou Tujia samples with the allele frequencies spanned from 0.0049 to 0.9709 and 0.0041 to 0.9675. For multi-copy loci, the number of alleles were 29 (DYS527a/b) to 50 (DYS385a/b) and 31 (DYS527a/b) to 44 (DYS385a/b) with the allele frequencies 0.0049–0.1214 and 0.0041–0.1423, respectively. GD values ranged from 0.0571 (DYS645) to 0.9665 (DYS385a/b) for Miao, and from 0.0638 (DYS645) to 0.9470 (DYF387S1) for Tujia. The HD and MP values of Miao were 1 and 0.00485, which of Guizhou Tujia were 0.999 and 0.00433. DC values of Miao and Tujia were computed as 100% and 98.37%. Moreover, the forensic parameters of the Yfiler and Yfiler Plus amplification systems (17 and 27 Y-STR loci) in the Guizhou Miao and Guizhou Tujia were separately calculated (table 1). In the Guizhou Miao population, the unique haplotype was 176 (Yfiler), 204 (Yfiler Plus) and 206 (AGCU Y37) with the HD of 0.9978, 0.9999 and 1. In the Tujia population, the unique haplotype was 211, 240 and 243 with the HD of 0.9978, 0.9998 and 0.999, respectively.

## 3.2. Comparison between Chinese and foreigners

To better understand the paternal genetic relationships between our studied populations and others, 26 different populations from home and abroad were obtained from YHRD. Pairwise genetic distances (Rst) calculated using AMOVA are shown in electronic supplementary material, table S5. The minimum genetic distance was between Yanbian Korean and South Korea Korean (0.0000), while the maximal was between Chamdo Tibetan and Laos Laotian (0.4223). For our studied populations, Guizhou Miao (0.0049) and Tujia (0.0041) had the closest genetic distance with Hunan Miao, while they both had the farthest genetic distance with Chamdo Tibetan (0.2432 of Miao and 0.2266 of Tujia).

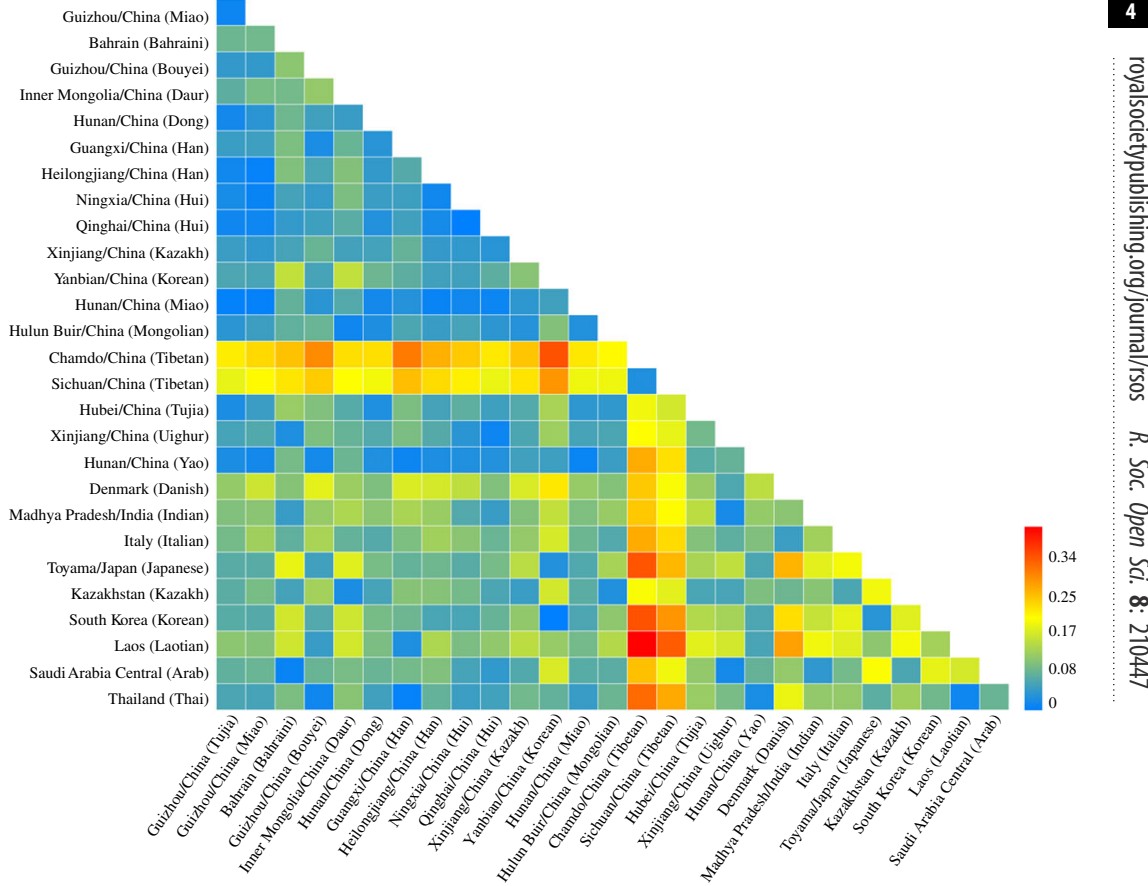

**Figure 1.** Heatmap of pairwise Rst values among the studied two populations and 26 reference populations.

**Table 1.** Standard forensic parameters based on 17, 27 and 37 Y-STR loci for the whole groups of Guizhou Miao and Guizhou Tujia populations.

| population | panel | HD | MP | DC | unique haplotype |
|---|---|---|---|---|---|
| Guizhou Miao | Yfiler-17 | 0.9978 | 0.0071 | 0.8544 | 176 |
| | YfilerPlus-27 | 0.9999 | 0.0049 | 0.9903 | 204 |
| | AGCU Y37 | 1.0000 | 0.0049 | 1.0000 | 206 |
| | Yfiler-17 | 0.9978 | 0.0062 | 0.8577 | 211 |
| Guizhou Tujia | YfilerPlus-27 | 0.9998 | 0.0043 | 0.9756 | 240 |
| | AGCU Y37 | 0.9999 | 0.0042 | 0.9878 | 242 |

The heatmap of pairwise Rst is shown in figure 1. With the gradual deepening of blue, yellow and red, the farther the genetic relationship among populations was. Guizhou Miao was close to Guizhou Tujia, Hunan Dong, Heilongjiang Han, Qinghai Hui, Ningxia Hui, Hunan Miao and Hunan Yao. Apart from Hulun Buir Mongolian and Hubei Tujia, Guizhou Tujia was approximately consistent with Guizhou Miao. Additionally, the studied populations were both far from the Tibetans (the flavescens colour).

As presented in the MDS plot (electronic supplementary material, figure S2), Hubei Tujia, Hunan Dong, Yanbian Korean and South Korea Korean were in the first quadrant; Chamdo Tibetan, Sichuan Tibetan, Kazakhstan Kazakh, Hulun Buir Mongolian and Inner Mongolia Daur were in the second quadrant; Xinjiang Kazakh, Italy Italian, Xinjiang Uighur, Bahrain Bahraini, Saudi Arabia Central Arab and Madhya Pradesh India Indian were in the third quadrant; Guizhou Miao, Hunan Miao,

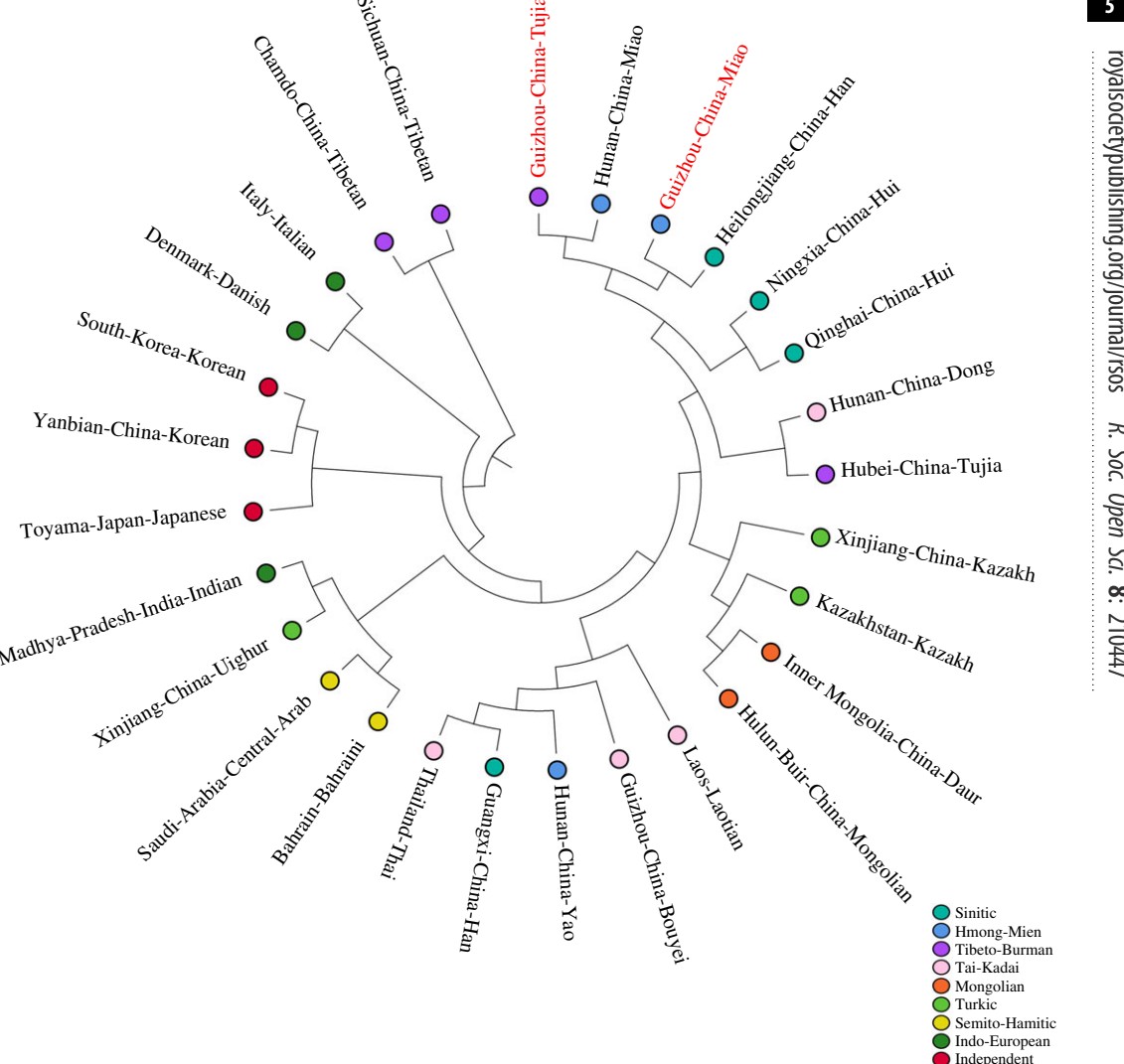

**Figure 2.** Neighbour-joining tree based on pairwise Rst values among 28 populations. (The red fonts are our studied populations.)

Hunan Yao, Qinghai Hui, Ningxia Hui, Guizhou Bouyei, Guangxi Han and two Southeast Asian countries (Laos Laotian and Thailand Thai) were in the fourth quadrant; Guizhou Tujia, Heilongjiang Han and Japan Japanese straddled the first and fourth quadrants. The Tibetans (Sichuan and Chamdo) located alone in the upper left corner, Xinjiang Uighur was close to West Asia (Saudi Arabia Arab and Bahrain Bahraini) and the European populations got together far from the Asian populations.

In the UPGMA tree (figure 2), there were three main branches. The first one combined all Asian populations (Esat Asia, Southeast Asia, South Asia, West Asia and Central Asia) except for the Tibetans; the second one contained two Europeans (Italy Italian and Denmark Danish); the third one included two Tibetans (Sichuan and Chamdo). Guizhou Tujia and Hunan Miao clustered at the same branch, Guizhou Miao and Heilongjiang Han clustered at the other same branch. Subsequently, the four populations grouped with other Asian populations. Additionally, the phylogenetic tree also revealed that populations tend to get together with the populations belonging to the same language family. For example, the Tai-Kadai family (Laos Laotian, Guizhou Bouyei and Thailand Thai), the Mongolian family (Inner Mongolia Daur and Hulun Buir Mongolian) and the Semito-Hamitic family (Bahrain Bahraini and Saudi Arabia Arab) gathered, respectively, and then clustered with others.

To further distinguish the genetic relationship among 28 populations, a PCA plot was employed (electronic supplementary material, figure S3). The PCA results were presented as the plots of the first three principal components (PCs). PC1 (63.34%) can differentiate the Asian populations from the European populations. Among them, the East Asian populations and the Southeast Asian populations cluster tightly. PC2 (22.27%) can separate the European related populations (Denmark Danish, Italy

Italian, Kazakhstan Kazakh and Xinjiang Uighur). PC3 (6.49%) can distinguish the Tibeto-Burman-speaking populations (Guizhou Tujia, Hubei Tujia, Sichuan Tibetan and Chamdo Tibetan) from the other populations.

# 4. Discussion

Presently, a novel 37 Y-STR loci system was tested in the Miao and Tujia populations residing in Guizhou Province. GD values of 37 Y-STRs were all higher than 0.5 with the exception of DYS391, DYS437, DYS438 and DYS645. Three multi-copy loci (DYS527a/b, DYS385a/b and DYF387S1) showed higher gene diversity than single-copy loci (GD > 0.9). Additionally, the AGCU Y37 PCR amplification kit includes all of Y-STRs in the previously developed forensic commercial kit (such as Yfiler and Yfiler Plus) and adds another seven low-medium mutation loci (DYS444, DYS447, DYS527, DYS557, DYS593, DYS596 and DYS645). More low-medium mutation loci increase the individual discrimination and play a vital role in forensic family research. And the additional new Y-STRs included in the AGCU Y37 significantly enhance the HD and DC, but decrease the MP. The results showed that 37 Y-STRs are highly polymorphic and informative. Thus, the AGCU Y37 amplification kit can be used in forensic for supplementary of autosomal chromosome STRs.

Furthermore, 27 Y-STRs included in the AGCU Y37 PCR amplification kit were used to explain the population genetic relationships. In the present study, we used multifarious bioinformatics methods (Rst genetic distance, heatmap, UPGMA, MDS and PCA) to reconstruct the population relationship of Guizhou Tujia and diverse ethnic groups from nine major language families coming from 11 countries (Sinitic: Han, Hui; Tai-Kadai: Bouyei, Dong, Laotian, Thai; Tibeto-Burman: Tibetan, Tujia; Hmong-Mien: Miao, Yao; Turkic: Uighur, Kazakh; Mogolian: Mogolian, Daur; Indo-European: Danish, Indian, Italian; Semito-Hamitic: Bahraini, Arab; Independent: Korean, Japanese). These bioinformatics methods generally employed the population comparisons in the forensic medicine [13–17]. The results showed that there were significant genetic differences between populations belonging to different regions and languages. Guizhou Miao and Tujia populations were all far from the Tibetans. In accordance with Zhang's research [18], the Tibetans differed from other East Asian populations because of the high-altitude adaptation genes (EPAS1 and EGLN1). Except the Tibetans, the East Asians were close to the Central Asians and the Southeast Asians. For the studied populations, Guizhou Miao and Guizhou Tujia had intimate relationships with Heilongjiang Han and Hunan Yao. The results of Rst genetic distance showed that Guizhou Miao and Tujia had the closest genetic distance with Hunan Miao (0.0049 of Guizhou Miao and 0.0041 of Guizhou Tujia) and Heilongjiang Han (0.006 of Guizhou Miao and 0.014 of Guizhou Tujia). In the MDS plot, although Guizhou Tujia and Heilongjiang Han straddled two quadrants, the two populations formed a cluster with Guizhou Miao and Hunan Miao. As for the consistent result of the four populations mentioned above using distinct methods, it might be explained by the geographical location and gene flows. First of all, Guizhou Tujia had an intimate relationship with geographically close Guizhou Miao populations and the Guizhou Han populations, which have a long history of living together and inter-mating. Secondly, the Guizhou Miao and Hunan Miao are of the same ethnicity living in adjacent provinces. Again, most Han minority dispersed in mainland China keeps intimate relationships, so the Guizhou Tujia showed a genetic affinity with Heilongjiang Han. Notably, Guangxi Han was far away from the studied populations, which was consistent with the previous studies [16,17]. Moreover, some differences were found between Northern and Southern China in the Chinese populations. The genetic relationship between the southern populations was tight while that between the northern populations was scattered. Additionally, our studied populations were close to the Sinitic-speaking populations. The results further reflected the gene exchange between the study population and the Chinese-speaking population were significant in a relatively close time period. Four analytical results were almost identical. These differences may probably be caused by geography, cultural, historical and linguistic factors. The same and similar ethnolinguistic and geographical cluster characteristics observed in our study were also be reported using other genetic markers, such as autosomal STRs and X chromosomal STRs (X-STRs). Take, for example, 15 autosomal STR loci [19] and 19 X-STRs [13] based on the same statistical methods in our previous phylogenetic relationship analysis; Guizhou Tujia had an intimate relationship with geographically close populations and the Han populations, whereas far distant from the Tibeto-Burman language-speaking populations. Totally, our findings based on the 37 Y-STRs demonstrate that Guizhou Miao and Guizhou Tujia are genetically similar with geographically close populations and other linguistically close populations, which is in accordance with the autosomal STR and X-STR consequences of geography and language classification.

# 5. Conclusion

We firstly reported the forensic parameters of 37 Y-STR loci from Miao and Tujia male individuals residing in Guizhou Province. This study followed the use of Y-STRs in forensic analysis under the recommendations of the International Society for Forensic Genetics (ISFG) [20]. All haplotype data were submitted to the YHRD and received accession numbers. These data showed high polymorphism and information in Guizhou Miao and Tujia populations. Additionally, the data could also provide support and supplement for forensic application and population structure.

Ethics. All of the volunteers had been adequately informed and signed the informed consent before sample collection. This study was approved by the Ethics Committee of the Zunyi Medical University (KLLY-2019-080). The procedures used in this study adhere to the tenets of the Declaration of Helsinki.

Data accessibility. All data are publicly available. Y-chromosomal data genotyped for this study have been submitted to the open access Y-STR Haplotype Reference Database (YHRD, www.yhrd.org) and are available under accession nos. YA004671 and YA004672. Additionally, the datasets supporting this article have been uploaded as the electronic supplementary material.

Authors' contribution. P.C., Y.B. and L.L. contributed to the study conception and design. Material preparation, data collection and analysis were performed by L.L. and L.Y., S.C., H.Z., M.L., X.H., J.Y. and C.L. The first draft of the manuscript was written by L.L. and L.Y., and all authors commented on previous versions of the manuscript. All authors read and approved the final manuscript.

Competing interests. The authors declared no competing financial interest.

Funding. This work was supported by a grant from the National Natural Science Fund of China (grant no. 81601651).

Acknowledgements. We have greatly appreciated the volunteers who contributed samples and Dr Dyke who edited the written English for our study.

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
