## [Peer Review File · Royal Society Open Science]

Review History

RSOS-210447.R0 (Original submission)

Review form: Reviewer 1

Is the manuscript scientifically sound in its present form?

Yes

Are the interpretations and conclusions justified by the results?

Yes

Is the language acceptable?

Yes

Do you have any ethical concerns with this paper?

No

Have you any concerns about statistical analyses in this paper?

No

Recommendation?

Accept with minor revision (please list in comments)

Comments to the Author(s)

Dear editor Lianne Parkhouse:

It is always a great honor to review a manuscript for Royal Society Open Science! I thank the contributors for their time and dedication preparing the submission "Forensic characteristics and population construction of 2 major minorities from Southwest China revealed by a novel 37 Y-STR loci system" (Submission ID RSOS-210447). In their study, the authors research the Forensic characteristics and population construction of Guizhou Miao and Guizhou Tujia using AGCU Y37 PCR amplification kit. The studied data could provide support and supplement for forensic application and population structure. But there are several problems need to be revised. The detailed comments are as follow:

1. In line 19-21 of Introduction, "Additionally, China legally recognizes 56 distinct ethnic groups with a population of around 1.4 billion in 2019" lacks references.
2. In line 22-24 of Introduction, "Miao and Tujia are the fourth and eighth largest minorities in China, which constitute approximately 0.7% and 0.6% of the total population" also lacks references.
3. The result of MDS plot shows that Guizhou Tujia and Heilongjiang Han are located in the first quadrant, and Guizhou Miao and Hunan Miao are located in the fourth quadrant. But the result of UPGMA tree shows that Guizhou Tujia and Hunan Miao clustered at the same branch, and Guizhou Miao and Heilongjiang Han clustered at the other same branch. The reason of the difference between MDS plot and UPGMA tree needs to be explained.
4. The supplementary table 5 requires a unified data format.

Review form: Reviewer 2

Is the manuscript scientifically sound in its present form?

Yes

Are the interpretations and conclusions justified by the results?

Yes

Is the language acceptable?

Yes

Do you have any ethical concerns with this paper?

Yes

Have you any concerns about statistical analyses in this paper?

Yes

Recommendation?

Accept with minor revision (please list in comments)

Comments to the Author(s)

The authors genotyped 37 Y-chromosome STR loci in two Chinese minorities (Miao and Tujia) residing in Guizhou province using the novel AGCU Y37 PCR amplification kit. The results generated in this study are reliable and meaningful for forensic practice and population studies with the data is highly polymorphic and informative. Population comparisons among 28 populations based on various methods and datasets showing valuable population relationships concerning to their ethnic, linguistic, or geographic origin. The manuscript should be considered to accept. However, some minor points that need the authors to address:

1. The amplification reactions and the thermal cycling parameters of the AGCU Y37 PCR amplification kit should be shown or supply a reference for the manufacturer's protocol for the AGCU Y37 PCR amplification kit.
2. The authors should do a careful read to minimize typographical, grammatical, and bibliographic errors.
3. Are the homogeneity and heterozygosity among Chinese populations revealed by Y-STRs consistent with the results shown by other genetic markers, such as autosomal-STRs or X-STRs? Are geographically or linguistically close populations clustered together? Please add some description and necessary discussion.
4. What is the basis for the author to choose the comparison group?

Decision letter (RSOS-210447.R0)

Dear Miss Luo

On behalf of the Editors, we are pleased to inform you that your Manuscript RSOS-210447 "Forensic characteristics and population construction of 2 major minorities from Southwest China revealed by a novel 37 Y-STR loci system" has been accepted for publication in Royal Society Open Science subject to minor revision in accordance with the referees' reports. Please find the referees' comments along with any feedback from the Editors below my signature.

Please submit your revised manuscript and required files (see below) no later than 7 days from today's (ie 21-Jun-2021) date. Note: the ScholarOne system will 'lock' if submission of the revision is attempted 7 or more days after the deadline. If you do not think you will be able to meet this deadline please contact the editorial office immediately.

on behalf of Professor Steve Brown (Subject Editor)
openscience@royalsociety.org

Reviewer comments to Author:

Reviewer: 1

Comments to the Author(s)

It is always a great honor to review a manuscript for Royal Society Open Science! I thank the contributors for their time and dedication preparing the submission "Forensic characteristics and population construction of 2 major minorities from Southwest China revealed by a novel 37 Y-STR loci system" (Submission ID RSOS-210447). In their study, the authors research the Forensic characteristics and population construction of Guizhou Miao and Guizhou Tujia using AGCU Y37 PCR amplification kit. The studied data could provide support and supplement for forensic application and population structure. But there are several problems need to be revised. The detailed comments are as follow:

1. In line 19-21 of Introduction, "Additionally, China legally recognizes 56 distinct ethnic groups with a population of around 1.4 billion in 2019" lacks references.
2. In line 22-24 of Introduction, "Miao and Tujia are the fourth and eighth largest minorities in China, which constitute approximately 0.7% and 0.6% of the total population" also lacks references.
3. The result of MDS plot shows that Guizhou Tujia and Heilongjiang Han are located in the first quadrant, and Guizhou Miao and Hunan Miao are located in the fourth quadrant. But the result of UPGMA tree shows that Guizhou Tujia and Hunan Miao clustered at the same branch, and Guizhou Miao and Heilongjiang Han clustered at the other same branch. The reason of the difference between MDS plot and UPGMA tree needs to be explained.
4. The supplementary table 5 requires a unified data format.

Reviewer: 2

Comments to the Author(s)

The authors genotyped 37 Y-chromosome STR loci in two Chinese minorities (Miao and Tujia) residing in Guizhou province using the novel AGCU Y37 PCR amplification kit. The results generated in this study are reliable and meaningful for forensic practice and population studies with the data is highly polymorphic and informative. Population comparisons among 28 populations based on various methods and datasets showing valuable population relationships concerning to their ethnic, linguistic, or geographic origin. The manuscript should be considered to accept. However, some minor points that need the authors to address:

1. The amplification reactions and the thermal cycling parameters of the AGCU Y37 PCR amplification kit should be shown or supply a reference for the manufacturer's protocol for the AGCU Y37 PCR amplification kit.
2. The authors should do a careful read to minimize typographical, grammatical, and bibliographic errors.
3. Are the homogeneity and heterozygosity among Chinese populations revealed by Y-STRs consistent with the results shown by other genetic markers, such as autosomal-STRs or X-STRs? Are geographically or linguistically close populations clustered together? Please add some description and necessary discussion.

4. What is the basis for the author to choose the comparison group?

===PREPARING YOUR MANUSCRIPT===

===PREPARING YOUR REVISION IN SCHOLARONE===

- 1) One version identifying all the changes that have been made (for instance, in coloured highlight, in bold text, or tracked changes);
 - 2) A 'clean' version of the new manuscript that incorporates the changes made, but does not highlight them.
 - An individual file of each figure (EPS or print-quality PDF preferred [either format should be produced directly from original creation package], or original software format).
 - An editable file of each table (.doc, .docx, .xls, .xlsx, or .csv).
 - An editable file of all figure and table captions.
- Note: you may upload the figure, table, and caption files in a single Zip folder.
- Any electronic supplementary material (ESM).
 - If you are requesting a discretionary waiver for the article processing charge, the waiver form must be included at this step.
 - If you are providing image files for potential cover images, please upload these at this step, and inform the editorial office you have done so. You must hold the copyright to any image provided.
 - A copy of your point-by-point response to referees and Editors. This will expedite the preparation of your proof.

- Ensure that your data access statement meets the requirements at <https://royalsociety.org/journals/authors/author-guidelines/#data>. You should ensure that you cite the dataset in your reference list. If you have deposited data etc in the Dryad repository, please only include the 'For publication' link at this stage. You should remove the 'For review' link.
- If you are requesting an article processing charge waiver, you must select the relevant waiver option (if requesting a discretionary waiver, the form should have been uploaded at Step 3 'File upload' above).
- If you have uploaded ESM files, please ensure you follow the guidance at <https://royalsociety.org/journals/authors/author-guidelines/#supplementary-material> to include a suitable title and informative caption. An example of appropriate titling and captioning may be found at [https://figshare.com/articles/Table_S2_from_Is_there_a_trade-off_between_peak_performance_and_performance_breadth_across_temperatures_for_aerobic_sc ope_in_teleost_fishes_/3843624](https://figshare.com/articles/Table_S2_from_Is_there_a_trade-off_between_peak_performance_and_performance_breadth_across_temperatures_for_aerobic_scope_in_teleost_fishes_/3843624).

Author's Response to Decision Letter for (RSOS-210447.R0)

See Appendix A.

Decision letter (RSOS-210447.R1)

Dear Miss Luo,

I am pleased to inform you that your manuscript entitled "Forensic characteristics and population construction of 2 major minorities from Southwest China revealed by a novel 37 Y-STR loci system" is now accepted for publication in Royal Society Open Science.

on behalf of Professor Steve Brown (Subject Editor)
openscience@royalsociety.org

Appendix A

Response to Reviewers

Dear Editors and Reviewers:

Thank you very much for giving us the opportunity to revise our manuscript. We also thank the anonymous Reviewers whose comments greatly improved this paper. According to the Reviewers' pertinent comments, we have made the required changes to the previous version. All the changes made to the original version are highlighted in red, and points by point responses to the Reviewers' comments are listed below. We hope that the revised version and our response have addressed the issues raised by the Editors and Reviewers and therefore have a better chance of being accepted by your journal.

In the following pages, we present our responses.

With many thanks and very best wishes,

Pengyu Chen

Corresponding author: Pengyu Chen

E-mail: pychenfs@163.com.

Affiliation: Key Laboratory of Cell Engineering in Guizhou Province, Affiliated hospital of Zunyi Medical University, Zunyi, 563099, Guizhou, PR China

Referee(s)' Comments

Reviewer #1:

1. In line 19-21 of Introduction, "Additionally, China legally recognizes 56 distinct ethnic groups with a population of around 1.4 billion in 2019" lacks references;

Response: Thank you very much for your valuable advice. We have added the official website address (https://en.jinzhao.wiki/wiki/China#cite_note-23) of the corresponding information resource. Up to now, the data has been updated to 2020 year. Therefore, we have changed the date in the text from 2019 to 2020.

2. In line 22-24 of Introduction, “Miao and Tujia are the fourth and eighth largest minorities in China, which constitute approximately 0.7% and 0.6% of the total population” also lacks references;

Response: Thanks for your comments. We have added the official website address (<http://www.stats.gov.cn/tjsj/pcsj/rkpc/6rp/indexch.htm>) of the corresponding information resource. This website recorded the population data of the sixth population census in 2010, including ethnic distribution, age distribution of China and so on.

3. The result of MDS plot shows that Guizhou Tujia and Heilongjiang Han are located in the first quadrant, and Guizhou Miao and Hunan Miao are located in the fourth quadrant. But the result of UPGMA tree shows that Guizhou Tujia and Hunan Miao clustered at the same branch, and Guizhou Miao and Heilongjiang Han clustered at the other same branch. The reason of the difference between MDS plot and UPGMA tree needs to be explained;

Response: Thanks for your suggestions. We are very sorry that we have made an inappropriate or even error description of the distribution of some main populations in the MDS plot (Supplementary Figure S2) in the Result part. Actually, the dots of Guizhou Tujia and Heilongjiang Han straddled the first and fourth quadrants which closely clustered with Guizhou Miao and Hunan Miao, and the four populations are distinctively intimate than others in the MDS Plot. Thereafter, the overall phylogenetic consistence was reflected between MDS plot and UPGMA tree.

However, the fine population relationship among various analysis methods still exists mainly in terms of their difference in basic theory and algorithm, so comprehensive data from multifarious population comparison rather than one single method were necessary to reconstruct population relationship.

As for the consistent result of the four populations mentioned above using distinct methods, it might be explained by the geographical location and gene flows. First of all, Guizhou Tujia had an intimate relationship with geographically close Guizhou Miao populations and the Guizhou Han populations, which have long history of living together and inter-mating. Secondly, the Guizhou Miao and Hunan Miao are of the same ethnicity living in adjacent provinces. Again, most Han minority dispersed in mainland China keeps intimate relationships, so the Guizhou Tujia showed genetic affinity with Heilongjiang Han. Similar phylogenetic relationships were also been demonstrated by other research using other

genetic markers such as autosomal STRs and X chromosomal STRs (see Discussions part)

4. The supplementary table 5 requires a unified data format;

Response: Thank you for your suggestion. We have provided the unified data format in a revised supplementary table S5, the decimal places were all set to four decimal places.

Reviewer #2:

1. The amplification reactions and the thermal cycling parameters of the AGCU Y37 PCR amplification kit should be shown or supply a reference for the manufacturer's protocol for the AGCU Y37 PCR amplification kit;

Response: Thank you. According to the Reviewer's suggestion, we have added the amplification reactions and the thermal cycling parameters of the AGCU Y37 PCR amplification kit in the "DNA amplification and STR genotyping".

2. The authors should do a careful read to minimize typographical, grammatical, and bibliographic errors;

Response: Thank you for your suggestion. We have carefully gone through the manuscript, corrected a few language errors. Additionally, the manuscript has been reviewed by Dr. Dyke, who worked in the University of Debrecen and revised based on his suggestions. The error revisions in the manuscript were marked in red.

3. Are the homogeneity and heterozygosity among Chinese populations revealed by Y-STRs consistent with the results shown by other genetic markers, such as autosomal-STRs or X-STRs? Are geographically or linguistically close populations clustered together? Please add some description and necessary discussion;

Response: Thank you. We compared the results of this study with those of previous studies on autosomal STR and X-STR loci, the results obtained agree approximately with those expected. Take, for example, 15 autosomal STR loci and 19 X-STRs based on the same statistical methods in our previous phylogenetic relationship analysis, Guizhou Tujia had an intimate relationship with geographically close populations and the Han populations, whereas far distant from the Tibeto-Burman language speaking populations. As common genetic markers in forensic medicine, these 3 genetic markers

all presented that genetic similarities are closely associated with language families and geographic regions. Additionally, we found that Guizhou Miao and Guizhou Tujia are genetically similar with geographically close populations and other linguistically close populations. We added some description and necessary discussion in the “Discussion”.

4. What is the basis for the author to choose the comparison group?

Response: Thank you. China has a vast territory and comprises 56 ethnic groups. In order to better clarify the relationship between our studied population and other populations, the reference populations mainly consider the following factors: (1) according to the China's geographical distribution, comparing as many groups as possible from different administrative divisions, languages families and ethnic groups throughout China reported based on the 27Y-STRs from YHRD database were selected for comparison; (2) the population data is reliable, which can represent the genetic structure characteristics of the reported population; (3) after the preference was given to Chinese groups, some foreign populations belonging different language families and continents near China as the reference populations were added.

Consequently, 28 populations belonging to 9 major language families from 11 countries were screened. Among them, 2 Hmong-Mien-speaking populations (Hunan Miao and Hunan Yao), 3 Tibeto-Burman-speaking populations (Hubei Tujia, Chamdo Tibetan, and Sichuan Tibetan), and the study populations belong to the same language family; 2 Sinitic-speaking populations (Guangxi Han and Heilongjiang Han) and 2 Tai-Kadai-speaking populations (Guizhou Bouyei and Hunan Dong) may have the same origin as the study populations or live near the study populations; 1 Independent-speaking population (Yanbian Korean), 2 Mogolian-speaking populations (Inner Mongolia Daur and HulunBuir Mongolian), 2 Sinitic-speaking populations (Ningxia Hui and Qinghai Hui), and 2 Turkic-speaking populations (Xinjiang Uighur and Xinjiang Kazakh) can be found in seven regions of China; 3 Indo-European-speaking populations (Denmark Danish, India Indian, and Italy Italian), 2 Independent-speaking populations (South Korea Korean and Japan Japanese), 2 Semito-Hamitic-speaking populations (Bahrain Bahraini and Saudi Arabia Central Arab), 2 Tai-Kadai-speaking populations (Laos-Laotian and Thailand-Thai), and 1 Turkic-speaking population (Kazakhstan Kazakh) coming from six continents were selected as the foreign reference populations.